



# Downstream ecosystem responses to middle reach regulation of river discharge in the Heihe River Basin, China

Y. Zhao[1], Y. P. Wei[1], S. B. Li[2], and B. F. Wu[3]

[1] School of Geography, Planning and Environmental Management, the University of Queensland, Brisbane, Australia 4072

[2] School of Geography and Remote Sensing, Nanjing University of Information Science and Technology, Nanjing, China 210044

[3] Division of Digital Agriculture & Disaster, Key Laboratory of Digital Earth Science, Institute of Remote sensing and digital earth, Chinese Academy of Sciences, Beijing, China 100101

*Correspondence to:* Y. Wei (yongping.wei@uq.edu.au)

**Abstract** Understanding the oasis ecosystem responses to upstream regulation is a challenge for catchment management for ecological restoration. This empirical study aimed to understand how oasis ecosystems including water, natural vegetation and cultivated land responded to the implementation of the Ecological Water Diversion Project (EWDP) in the Heihe River in China. The annual Landsat images from 1987 to 2015 were firstly used to characterize the spatial extent, frequency index and fractional coverage (for vegetation only) of these three oasis ecosystems and their relationships with hydrological (river discharge) and climatic variables (regional temperature and precipitation) were explored with linear regression models. The results show that river regulation of the middle reaches identified by the discharge allocation to the downstream basin experiences three stages, namely decreasing inflow (1987–1999), increasing inflow (2000–2007) and relative stable inflow (2008–2015). Both the current and previous years' combined inflow determines the surface area of the terminal lake ($R^2 = 0.841$). Temperature has the most significant role in determining broad vegetation distribution, whereas hydrological variables had a significant effect only in near-river-channel regions. Agricultural development since the execution of the EWDP might have curtailed further vegetation recovery. These findings are important for the catchment managers' decisions about future water allocation plans.



## 1 Introduction

Rapid population growth and economic development over the past several decades have led to overuse of water resources and serious ecological degradation of water catchments worldwide. In arid and semiarid regions where water is very scarce, oasis ecosystems in downstream floodplains are particularly threatened by increasing upstream water diversion for socio-economic development (Cochrane et al., 2014; Lu et al., 2015). Understanding the oasis ecosystem responses to upstream regulation is a challenge for catchment managers who wish to implement ecological restoration efforts.

Downstream ecosystem responses to regulation of upstream river discharge are complex processes that are influenced by many factors. Oasis vegetation, the dominant component in the ecosystem, is sensitive to water availability, which is in turn substantially influenced by regional temperature, precipitation and hydrological variations. Therefore, vegetation changes, along with presence of water bodies (Lu et al., 2011), are important indicators for understanding oasis ecosystem changes. There have been a large number of studies on understanding regional climatic and hydrological variations and their interactions with vegetation which are mainly aimed to improve the simulation of discharge generation. Such studies are based on the analysis of hydrologic alterations and the relationship with regional climate variability (Chen et al., 2007; Seyedabbasi et al., 2011); or on vegetation dynamics at catchment scale (Tang et al., 2012; Tesemma et al., 2014); or on human-induced land use change and irrigation (Thanapakpawin et al., 2007; Yu et al., 2015). An increasing number of studies on water allocation for human use and the environment have been conducted to define optimal river flows to sustain ecosystem integrity (Bunn and Angela, 2002; Pahl-Wostl et al., 2013; Wang et al., 2015), and to identify trade-off relationships between competing economic and environmental goals (Cheng et al., 2014; Hu et al., 2015a; Schlüter et al., 2005; Wang et al., 2007), and between the upstream and downstream areas of river basins (Barbier, 2003; Lu et al., 2015; Thevs et al., 2015). However, the impact of upstream river regulation on oasis ecosystems has received little attention. Considering its importance, it is necessary to determine the interactions between upstream regulation and downstream ecosystem responses. This requires systematically analysis of temporal variations in upstream water-use records, as well as downstream oasis development.

An oasis is a specific ecosystem that exists within arid and semi-arid catchments. Ejina Oasis is a typical oasis system that has suffered great environmental degradation due to increased water scarcity and human intervention. With its flat land, adequate sunlight and sufficient water sources coming down from the Qilian Mountains, the Ejina Basin, together with the Hexi Corridor lies in middle reach, has been an important grain production region of China, which could be traced back over 2000 years. Since the establishment of new China government in 1949, the Basin has experienced water and ecological stress. Increased water withdrawals for agricultural irrigation and municipal water supplies in the Hexi Corridor have significantly reduced the river flows to the lower reaches, which threatened large areas of woodland and natural oasis in the Ejina Basin and created visible signs of ecological degeneration and desertification (Zhu et al., 2009). The decreased water flows into the





terminal lakes caused the West Juyan Lake to dry up in 1961 and the East Juyan Lake in 1992. Until the late 1990s, the Chinese Government implemented a series of policies, including converting farmland back to forest and grass and implementing Ecological Water Diversion Projects (EWDPs), to ensure the delivery of minimum amount of water to the lower reaches of the Basin for the mitigation of ecosystem degradation in the region. Specifically, water allocated to the downstream region should

be over 950 million m$^3$ once the streamflow at Yingluoxia exceeds 1,580 million m$^3$ according to the EWDP implemented in 2000, only about 630 million m$^3$ could be retained for middle reach use. However, direct water diversion from Heihe River in 2000 was reported to be 840 million m$^3$ (Shi et al., 2014). The competing demands between middle reach consumption and downstream environmental use lasted after execution of the EWDP and restoring the downstream ecosystem remained challenging the Basin managers. The trajectory of river regulation and Basin development provided an ideal model to study

arid oasis changes in response to water exploitation activities.

    Long-term datasets are essential in determining environmental changes and assessing rehabilitation efforts in regulated river basins. Along with the available long-term hydrological and climatic records for the study area, we utilized Landsat images with 30 m spatial resolution for the period 1987–2015. To our knowledge, this is the first attempt to apply high-resolution images and long term datasets in Ejina Oasis. Several previous studies have applied MODIS datasets, available from 2000, to

detect temporal variations in vegetation in similar arid ecosystems (Hu et al., 2015a; Hu et al., 2015b; Jia et al., 2011). Although MODIS has good temporal resolution, its coarse spatial resolution may not allow accurate detection of small objects and capture fine-scale details (Fensholt et al., 2009), especially in the arid ecosystem of the Ejina Basin where the vegetation is mostly sparsely distributed and major changes only occur 100 – 400 m away from the water channels (Guo et al., 2008). The 30 m Landsat images used in this study enable the capture of finer details and the better identification of landscape changes,

and therefore help the understanding of the underlying causative factors.

    The primary goal of this study was to develop an understanding of the downstream oasis ecosystem responses to middle reach river regulation in the Ejina Basin over the past 30 years, during which significant alterations in water policies and human intervention have occurred. The specific objectives were to: (1) determine streamflow changes due to the regulation for human water use in the middle reaches; (2) characterize the downstream ecosystems using 30 m resolution Landsat images; (3)

determine the relationship between the downstream ecosystem change and middle reach river regulation in the context of hydrological changes in both the middle reaches and further downstream. It is expected that the findings from this study will help understanding of how the Basin ecosystem responds to water policy changes, which has important implications for future management actions.



## 2 Methods

### 2.1. Study area

The Ejina Oasis (99.7° – 101.7° E and 40.45° – 42.5° N) is located in the lower reaches of the Heihe River Basin (HRB), the second largest inland river basin of China, in the Inner Mongolia Autonomous Region (Figure 1). This area is part of the Gobi

Desert with a mean elevation around 1,000 m. The region has a typical continental arid climate, with an annual average temperature of 8.8°C over the last five decades. Annual precipitation in this region is scarce, averaging 35 mm over the past 50 years, but varying widely from 7 to 101 mm, whereas annual potential evaporation reaches 2,300 mm. With such a dry climate, surface runoff is rare in this region and therefore discharge down the Heihe River is the major water resource for local economic development. The headwaters of the rivers flowing into the Basin are in the Qilian Mountain, which flow to the

middle reach plains between Yingluoxia (YLX) and Zhengyixia (ZYX) Gauge Stations. The downstream oasis is fed by the Heihe River and its tributaries, the most important of which are the Donghe River and Xihe River (Figure 1).

(Figure 1)

### 2.2. Hydrological and climatic variables influencing the downstream ecosystems

This study focused on discharge at YLX and ZYX Gauge Stations to gain insights into hydrological variations, as well as the

impacts of water regulation in middle reach catchments. The YLX station is located at the junction between upper and middle reaches of the HRB and represents the major water resource for sustaining consumption both in middle reach and downstream catchments. ZYX is located at the junction of middle and lower HRB, and represents the proportion of water allocated for downstream developments. The absolute value of streamflow at ZYX ($Qzyx$), the streamflow consumed in the middle reaches ($\Delta Q$, the difference in streamflow between YLX and ZYX), and the percentage of water allocated for the downstream basin

Rzyx ($Qzyx/Qylx$) were used to understand the interactions between the middle reach regulation and the consequential changes in downstream hydrological conditions. Monthly discharge records between 1987 and 2014 at the YLX and ZYX stations were collected from WestDC (http://westdc.westgis.ac.cn/) maintained by the Cold and Arid Regions Environmental and Engineering Research Institute of Chinese Academy of Sciences. Discharge records for the 1960s were also included as baseline values, reflecting discharge levels with little human intervention in the HRB.

Besides the streamflow allocated to downstream regions, precipitation and temperature were selected as two major climatic variables to understand vegetation dynamics in this arid region. Daily precipitation records at Ejina Meteorological Station from 1987 were collected from WestDC to calculate the annual precipitation. Daily mean temperature, which accompanies this dataset, was also processed to obtain annual mean temperature to assist further analysis.

### 2.3. Indicators for determining downstream ecosystem

The Ejina Basin is an ecosystem consisting of a Gobi Desert zone, vegetation, water-covered regions and cultivated lands. The



Gobi Desert zones occupy about 90% of the region and, owing to the extremely low precipitation, these areas are vulnerable to any climatic changes, especially the regions further away from the river channels where less affected by the river discharge. Therefore, this study focused on the native vegetation, water-covered regions and cultivated land. Native vegetation is import to maintain series of ecosystem functions and services, including soil/water/nutrient regulation, biological control, wildlife habitats and providing food and water for local people. Local wetlands are vital to support the survival of surrounding vegetation and habitats in this arid region. Cultivated land in this study was represented by vegetated areas resulting from human intervention. Although not a natural ecosystem, it can be used to consider the impact of human activities on the vegetation dynamics.

Table 1 shows the metrics used to characterize the above three components. The spatial extent of vegetation, water-covered regions and cultivated lands were derived through image classification to identify distributions under various level of water availability. The frequency index for vegetation and water covered regions was introduced to illustrate their spatial distribution during three identified periods with different flow regimes and to follow their changes between the periods. The three periods were defined according to a preliminary inspection of temporal variations in streamflow entering the downstream catchment, namely: steadily decreasing (1987–1999), variably increasing (2000–2007) and relative stable (2008–2015). For vegetation-covered regions only, fractional vegetation cover (FVC) was calculated using a linear mixture model.

To show the potential influence of different water availability in near-river to distant regions, a series of buffer zones along river channels (100 m, 400 m and 1000 m away from river channel) were introduced to detect the interactions of vegetation dynamics and river flow. These three buffers were determined from previous work illustrating that vegetation grew well within 400 m of the water channels, while the growth almost ceased beyond 1000 m from the channels (Guo et al., 2008). Areas of vegetation-covered regions and the corresponding mean FVC values for the selected buffer zones were extracted for further analysis.

(Table 1)

### 2.4. Landsat images collection, preprocessing, interpretation and validation

Landsat Thematic Mapper (TM) or Operational Land Imager (OLI) imageries (hereafter simply referred as "Landsat images") were adopted to derive the selected metrics. Processing of the datasets were detailed in the following sections.

#### 2.4.1. Landsat images collection and preprocessing

One Landsat scene (path/row of 134/31) is required to cover the study area. We acquired annual images for most years from 1987 to 2015 except for 1989 and 1997 (27 scenes in total). The cloud-free images were mainly taken during June to October (except data in 1993 acquired in February), which would represent the growing season for crops, grasses and forests with relative high coverage. This period also covered the major water allocation events within the HRB and could help to determine





the maximum water surfaces. The images were downloaded through USGS Earthexplorer (http://earthexplorer.usgs.gov/).

Digital numbers (DN) of the Level-1T products were converted into top-of-atmosphere (TOA) reflectance using the radiometric gain and offset values associated with each image. Then we adopted a QUick Atmosphere Correction (QUAC) method to account for atmospheric scattering and deriving land surface reflectance (Bernstein et al., 2012). Since the

differences in acquisition date might also contribute to variations in spectral signals due to different atmosphere and ground conditions, we corrected by normalizing all other images to a selected strictly cloud-free scene to minimize the impact. The 2001 scene (DOY 224) was set as reference, a group of invariant pixels were selected for each image pair (2001 and an extra year) and the relative normalization was performed through linear regression analysis based on the selected pixel values. The calibration and normalization procedures were programmed and debugged with Interactive Data Language (IDL 8.2.3).

**2.4.2.     Image interpretation for determining the selected metrics**

Water and vegetation (including crops) distributions were determined through an unsupervised classification approach. Each image was separately classified using the ArcGIS (version 10.2) Iso cluster unsupervised classification procedure to derive 20 spectral clusters. The clusters were then assigned to three broad subclasses: (1) vegetation (crops, forests and grass land); (2) water surface (inundated river channels, ponds and terminal lakes); and (3) bare surfaces (Gobi Desert and residential areas)

based on their spectral similarity and surrounding clusters. Because the landscape composition in the region was relative simple, this classification procedure could effectively determine the three subclasses with the spectral information of the multispectral images.

We then applied a knowledge-based classification method to deduct the cultivated lands from the classification result. Briefly, with the most recent available high-resolution Bing Aerial maps (mostly in 2013), we firstly created a vector layer that

maximum reflected cultivated land distribution (including the lands under operation, fallow and abandoned croplands) by digitizing on-screen patches through shape and texture characters. The Normalized Differential Vegetation Indices (NDVI) were then calculated using red and near infrared bands for each year and a threshold analysis was applied to derive possible cultivated lands. The results were then overlaid with the created vector layer and each corresponding Landsat image to determine the final cultivated land distribution for each year. The derived cultivated lands were then deducted from the previous

obtained vegetation results for native vegetation distribution maps.

The per pixel frequency indices for water and vegetation were then calculated through counting the times with water or vegetation distribution during each period and divided by eight (length of each period). For the first period, classification results of 1987 and 1999 were neglected in this calculation to assure the consistency of the results (no data available for 1989 and 1997, result in 1993 with poor quality excluded). The change in frequency index between the periods was calculated by

subtracting from one another and the obtained change rates were further grouped into six levels to reflect minor, moderate and significant changes. The levels were determined based on standards from Thomas et al. (2011): significant decrease (–1 to –





0.67), moderate decrease (–0.67 to –0.33), minor decrease (-033 to 0), minor increase (0 to 0.33), moderate increase (0.33 to 0.67) and significant increase (0.67 to 1).

The FVC values of each pixel were calculated using a commonly used linear mixture model (Carlson and Ripley, 1997; Zeng et al., 2000). The model was described as follows:

$$\text{FVC} = (NDVI - NDVI_{soil})/(NDVI_{veg} - NDVI_{soil})$$

where $NDVI_{veg}$ is the NDVI value of fully vegetation covered pixels, and $NDVI_{soil}$ is the value of bare soil pixels in the image. To determine $NDVI_{veg}$ and $NDVI_{soil}$, we calculated the accumulation percentage of the NDVI values for each year and chose the $NDVI_{veg}$ and $NDVI_{soil}$ values according to the following: the studied region largely comprised with Gobi Desert, therefore we set the NDVI at 20% of pixels as $NDVI_{soil}$; while there were very limited "pure" vegetation covered pixels in this arid

region, we then set the value of the last 100 pixel as $NDVI_{veg}$.

### 2.4.3.    Validation of the derived metrics

In this study, we used the high-resolution satellite and aerial imageries from the Bing Aerial Maps service of ArcGIS online to validate the water and vegetation classification results. Bing Aerial Maps offers a series of orthographic aerial and satellite imagery from 1999 onwards at different spatial resolutions (from 30 m to < 1 m). The aerial map was loaded in ArcGIS and

the acquisition date of the displayed imagery was determined with the Bing Aerial Imagery Analyzer for OSM (http://mvexel.dev.openstreetmap.org/bing/). Then the classification result for each year was visually checked, the vegetation and water patches that failed to be characterized by the unsupervised classification were manually digitized and added to the image, while the wrongly classified patches were deleted. For all other years, a visual comparison of the class distribution was made with each corresponding Landsat image (Thomas et al., 2011). Classification anomalies (cloud contamination, mixed

class pixels, and so on) were removed manually.

Since it was not practical to use the limited available high-resolution imagery to validate the time series FVC results, we applied a compromised method to assess the derived values in the desert, where FVC was supposed to be 0. Specifically, a layer with 1,000 randomly selected sampling points distributed in absolute non-vegetation deserts was established. FVC values for each year were extracted and analyzed. Mean FVC for the 1,000 randomly selected points ranged from –3% to 1.3% and

annual variation was less than 2% for most years. Therefore, we could infer that the selected $NDVI_{soil}$ and $NDVI_{veg}$ for each year were rational and the calculated FVC was capable of reflecting long-term changes in vegetation developments.

### 2.5.    Determination of the response of different downstream ecosystem components to middle reach river regulation in the context of hydrological changes

Linear stepwise regression models were applied to link the changes in spatial extent of vegetation, surface water, and derived

FVC values, with regional climatic variables along with detected trends in hydrological variables. The regression procedure can select the best-fit combination of independent variables for dependent variable prediction with forward-adding and





backward-deleting variables using a critical F value to check the eligibility of the step forward added variables (Chen et al., 2013). A general form of the model can be described as:

$$y_i = \beta_1 x_{i1} + \cdots + \beta_n x_{in} + \varepsilon_i = X_i^T \beta + \varepsilon_i$$

where $T$ denotes transpose; $y$ is the dependent variable; subscript $i$ denotes for vegetation area, lake surface area and cultivated

land area in this study; $x$ represents the independent variables including current year's discharge at ZYX ($Q_c$), previous year's annual discharge ($Q_p$), total discharge of current year and previous years' ($Q_{cp}$), regional average temperature and precipitation; $\beta$ is a regression coefficient, where a positive $\beta i$ indicates positive correlation and vice versa.

Two-tailed Pearson correlation was also introduced to test the relationship between total vegetation dynamics (in spatial extent and FVC) and above-mentioned variables within the selected buffer zones.

## 3    Results

### 3.1.  Impact of middle reach river regulation on the streamflow downstream

The streamflow at YLX and ZYX (Figure 2) showed a synchronous increasing or decreasing trend for most years during the study period, which indicated that water flow in the Heihe River followed the regulation scheme. Overall, annual river streamflow at both YLX and ZYX has increased for the past three decades. However, the annual streamflow at ZYX decreased

slightly before 2000, followed by a rapid increase from 2000 to 2007 and has been relative stable since 2008. Specifically, annual streamflow at YLX has increased by 40% from about $15.74 \times 10^8 \, \mathrm{m}^3$ in 1987 to $22.03 \times 10^8 \, \mathrm{m}^3$ in 2014. The average annual streamflow at YLX for the study period (1987–2014) was $17.26 \times 10^8 \, \mathrm{m}^3$, which also increased by 17% when compare with the level in the 1960s ($14.71 \times 10^8 \, \mathrm{m}^3$). Meanwhile, annual streamflow at ZYX also increased by 34% from 1987 to 2014, whereas the average annual streamflow has decreased by 10% since the 1960s: from $10.66 \times 10^8 \, \mathrm{m}^3$ in the 1960s to $9.63 \times 10^8$

$\mathrm{m}^3$ in the study period. Water consumption in the middle reaches contributed most to this diverse variation.

The water diverted from the middle reaches ($\Delta Q$) increased from about $6.72 \times 10^8 \, \mathrm{m}^3$ in the 1980s to $8.11 \times 10^8 \, \mathrm{m}^3$ in the 1990s. A substantial decrease to about $6.67 \times 10^8 \, \mathrm{m}^3$ was observed in the early 2000s and this level remained fairly constant, until a slight uptrend started around 2007, reaching about $7.98 \times 10^8 \, \mathrm{m}^3$ in the 2010s. Accordingly, the ratio of streamflow allocated to Ejina Basin ($Qzyx/Qylx$) also underwent three stage changes: an initial decrease with variations (1987–1999), a

substantial increase (2000–2007) and then relative stable (2008–2015).

(Figure 2)

### 3.2.  Temporal changes of the downstream ecosystem indicators

Vegetation distribution within Ejina Oasis between 1987 and 2015 was highly variable, ranging from 528 to 1,025 km². From 1987 to 1999, there was a decreasing trend, as shown by the 5-year moving average, where annual vegetation reduced by



~30%, from about 882.7 km$^2$ in early 1990s to about 619.3 km$^2$. For the period from 2000 to 2007, the region showed relative stable vegetation distribution, averaging about 701.5 km$^2$. Since 2007, Ejina has experienced a steady upward trend in vegetation distribution, with an annual increase of 26.4 km$^2$ (Figure 3).

Within the vegetation covered regions, the overall mean FVC was 28% between 1987 and 2015, ranging from 22% to 35%. The changes in FVC over time shows a two-stage development, rather than three stages as previously identified. A significant downward trend occurs until 2003, with an annual decrease of 0.3%. After 2003, a non-statistically significant upward trend was observed, with a 0.2% increase per year ($R^2 = 0.379$).

(Figure 3)

Landsat images showed that East Juyan lake had dried up for most of the years between 1987 and 2001 (except 1988, 1989, 1993 and 1998, which had relatively high discharge events, Figure 3). Streamflow at ZYX for these years were $10.56 \times 10^8$ m$^3$, $15.74 \times 10^8$ m$^3$, $10.41 \times 10^8$ m$^3$ and $11.20 \times 10^8$ m$^3$, respectively, while for the other years before 2000, the average discharge was only $7.55 \times 10^8$ m$^3$. Since 2002, increased discharge has caused steady flows into East Juyan Lake and the lake surface increased from about 15.14 km$^2$ in 2002 to 45.73 km$^2$ in 2009. Thereafter, the average lake surface area was about 39 km$^2$, with little variation. The situation was similar for the overall total surface water, which also included the inundated river channels.

Cultivated lands in the Ejina Oasis experienced constant expansion (Figure 3), even during the period of increased water stress, as demonstrated by decreasing river discharge at ZYX (1987–1999). A more dramatic rise was observed between 2000 and 2006, by which time the cultivated land has almost doubled in area to about 84.3 km$^2$. Since 2007, the rate of increase rate has been very low, with the average total area of cultivated lands at 87.5 km$^2$, with little inter-annual variation.

### 3.3. Relationship between the downstream ecosystem changes and the streamflow

The linear models developed to demonstrate the relationship between the change of area of vegetation, lake surface and cultivated land in the downstream Ejina Oasis and hydrological and climatic variables are listed in Table 2. None of the hydrological variables showed significant influence on annual vegetation distribution, but regional temperature exerted significant negative effects, and it alone could explain about 52.6% of the variations in the area of vegetation. No significant relationship was obtained between mean FVC and hydrological or climatic variables. The linear models for Juyan Lake indicates that area of the lake surface was closely dependent on the combined previous and current year's discharge to the downstream basin. Compare with the significant positive effect of previous year's discharge ($R^2 = 0.365$), the combined river discharge of the current plus previous year showed a more robust relationship ($R^2 = 0.841$), which might indicate the importance of accumulated river flows. Agricultural activity relies on water, and this is demonstrated in this region by a significant positive relationship between current river discharge and detected cultivated lands ($R^2 = 0.530$).

(Table 2)



### 3.4. Evidence from spatial variations of ecosystem indicators

As indicated with the per pixel frequency index, most of the vegetation in this area was concentrated in the entry side of the Oasis and within the core Oasis area located in lower Donghe River (Figure 4). Although fed with increasing river discharge in period 2 (2000–2007), there was still obvious signs of decreased frequency of vegetation distribution for most of the areas

when compared to period 1 (1987–2009). While for period 3 (2008–2015), increased frequency was observed, especially along Xihe River and in the south part of the core oasis area. More specifically, from period 1 to 2, about 74% of the oasis vegetation regions experienced some kind of decrease (minor 47%, moderate 24% and significant 3%). Most moderate decrease occurred in the downstream reaches of the Donghe River, the upper reaches of the Nalin River and the entrance of the Donghe River (Figure 4 (a, d)). Only 21% of the regions showed minor increases, which were mainly distributed along the river channels.

From period 2 to 3, over 75% of the regions showed increased frequency (Figure 4 (b, e)). About 30% of the regions show moderate to significant increase, which were mainly distributed along Xihe River and its branches, around the Juyan Lake and outer circle of the core area of the Ejina Oasis. Around 20% of the regions experienced minor decreases during this period, which were major distributed along the Donghe River and within the core area of the oasis (Figure 4 (e)). For the whole study period (period 1 to 3), about half (48%) of the oasis regions experienced decreased frequency of vegetation distribution, which

were mainly distributed along Nalin River and within the core oasis area (Figure 4 (f)). Meanwhile, regions with increased frequency (52%) were mainly distributed along the Xihe River, around the Juyan Lake and in the southern part of the core area of the Ejina Oasis (Figure 4 (f)).

(Figure 4)

Only one or two inundation events along Xihe River were detected in period 1(Figure 5 (a)). After the water reform in 2000,

the frequency did not show obvious increase but broader regions were inundated for two to three times in this area (Figure 5 (b, c)). Meanwhile, the Donghe River and Juyan Lake experienced more frequent inundation, especially during period 3. The difference in frequency index between period 1 and period 3 also revealed that, during the whole study period, most of the increased water inundation was concentrated along the Donghe River channel and east Juyan Lake (Figure 5 (d)). The frequency maps also indicated that the Donghe River was not connected with the terminal lake for most years, which could be

attributed to two possible reasons. The first is that the width of the streams in lower river courses was below the detection limit of Landsat, while the second is that water did not flow constantly into the terminal lake, but depended on several drainage events that were not captured by the Landsat images.

(Figure 5)

Detailed correlation analysis within the buffer zones also demonstrated different levels of response along the Xihe and

Donghe Rivers. Specifically, while temperature showed a similar negative relationship with vegetation distribution in all three buffer zones, the correlation coefficient increased as the distance to the river channel increased. Although none of the





hydrological variables (current and previous years' total discharge at ZYX) showed any impact on total vegetation distribution in the Ejina Oasis, the previous 3 to 5 years' total discharge showed significant positive effects over the vegetation expansion in the selected buffer areas, especially along Donghe River. Vegetation in near-river regions showed more significant correlation with the previous year's discharge (Table 3).

5    (Table 3)

As for FVC within the buffer zones, temperature showed no impacts, but the previous 3–5 years' water discharge at ZYX showed significant positive correlation with the FVC values. According to the correlation coefficient detailed in Table 3, vegetation distributed along Donghe River and its tributaries seemed to be more responsive to flow variations. Furthermore, the mean FVC in different buffer zones showed similar decreasing trends before 2003 but with different rates: in near river

10    regions (within 100 m) the decreasing rate was -0.09% per year while this rate rise as distance to the river channel increased, and the decreasing rates in all buffer zones along Xihe River were about 0.1% higher than those along Donghe River (Table 4). After 2003, an upward trend (rate > 0.2% per year) was observed within the buffer zones and vegetation along Donghe River showed about a 0.2% higher increasing rate than that along the Xihe River (Table 4). Near-river-channel vegetation with higher increasing rates were also observed along both Xihe River and Donghe River.

15    (Table 4)

4    **Discussion and conclusions**

This paper comprises an empirical study on downstream oasis ecosystem responses to water regulation in the middle reach catchment in the HRB over the past 30 years. Changes in vegetation distribution, water bodies and cultivated lands were quantified using 30 m Landsat imagery from 1987 to 2015. Linear models were established to understand the impacts of middle

20    reach regulation of streamflow on downstream oasis ecosystems. The major research findings and their implications for future research and water management practice are described below.

This study revealed the controlling role of streamflow at ZYX station on downstream water availability and its response to the middle reach regulation. Water consumption in the middle reaches of the HRB has steadily increased since the early 1980s and depleted the amount of water allocated to ZYX. Consequently, most river channels and lakes in Ejina Basin experienced

25    very few inundations before 2000, especially along Xihe River and lower Donghe River. After 2000, consumption in middle reaches was regulated to meet the objective of downstream ecological water requirements (minimum of $9.5 \times 10^8$ m$^3$). Broader regions were subjected to inundation after the execution of EWDP. The situation was similar for the East Juyan Lake, which dried up in 1992, but was re-inundated in 2002 and has been subjected to constant inundation since that time. It is noteworthy that during 1992–2002 there were particular years with high discharge events (> $9.5 \times 10^8$ m$^3$), but the lake system did not





show immediate sign of recovery. With this knowledge, we concluded that repeated inflows in consecutive years are required to push water into the terminal lake, and hence support the surrounding environment: it not simply a matter of setting a certain level of water allocation. This was further supported by the significantly higher correlation between lake surface area and total discharge of the current and previous years combined.

5  Although synchronous trends were observed, water allocated to the Ejina Basin did not have a statistically significant effect on overall vegetation development. None of the hydrological variables we tested showed a statistical significant contribution to total vegetation distribution within the Ejina Oasis. Instead, the temperature seems to determine the spatial extent of vegetation in this region. Considering the extremely dry conditions in the region, increasing temperature would substantially deplete the already limited water supply through evaporation. This would ultimately restrict vegetation growth and expansion,

10 especially in the regions away from river channels, where the groundwater is too deep to sustain vegetation growth (Hu et al., 2007). Observations in this area (data not presented here) have found that the annual average groundwater depth alongside the river (< 300 m) was around 2 m, which increased to more than 3.5 m in remote regions (> 4300 m away from river channels). Water discharges would cause fluctuations in the groundwater depth only in the near-river regions (< 300 m) and would have no effect further away. This explains why the only significant impacts of hydrologic variations on vegetation was observed

15 within the selected buffer zones. Within these near-river regions, total streamflow from the previous 3-5 years had a positive impact on vegetation distribution, although temperature was still an important contributing factor. The time lag between streamflow changes and vegetation responses could be attributed both to inherent phenology processes and the time required for groundwater recharge (Jia et al., 2011; Wen et al., 2005). This point is also supported by the moderate to significant drop in frequency of vegetation distribution during 1987 to 2007 mainly along the Nalin River and lower reaches of Xihe and

20 Donghe Rivers (Figure 4), where the river channel was rarely inundated or the depleted streamflow was insufficient in the lower river reaches (Figure 5). Such conditions could have restricted the recharge process for the shallow aquifers within the region.

  Because vegetation does not respond quickly to streamflow changes, short-term regulations to increase water allocation to downstream basin for particular years may fail to support ecosystem development. Before the implementation of the EWDP

25 in 2000, there were years with high streamflow at ZYX, vegetation did not show signs of recovery but continued to recede because of the overall water scarcity (mean streamflow at $8.4 \times 10^8$ m$^3$ with a decreasing trend). The situation did not change until the mean streamflow reached $\sim 10.6 \times 10^8$ m$^3$ in early 2000s, with little annual variation, when the vegetation stabilized. A higher allocation for ecosystem services is required to support sustained vegetation recovery in future. The different responses of FVC to changes in water availability in the Xihe and Donghe Rivers provided further evidence for this point. The

30 Donghe River experienced more frequent inundation, and therefore had higher water availability, and significant correlation between FVC changes and streamflow variations only detected in this river. Trend analysis also found that, when compared





with Xihe River, vegetation along Donghe River showed lower FVC decreasing rate under water scarce conditions before the water reform and presented higher recovering rates after the EWDP was fully in operation (since 2002)

The attenuated effect of allocated water over vegetation distribution could be also attributed to agricultural development within the Ejina Basin. Although small in scale, agriculture within Ejina Basin still acts as a major competitor for water with

surrounding vegetation in this sparsely populated arid area. Before execution of EWDP, expansion of agriculture in Ejina Basin was slow and concentrated in a much smaller range than the oasis regions with decreasing vegetation distribution (Figure 4), which indicated few local human interventions during this period. However, over the whole study period, the regions with decreasing natural vegetation cover corresponded closely with the expansion of cultivated land, which shows that claiming land for agricultural activities causes the oasis vegetation to recede in these areas. Moreover, the spatial correlation between

expanding cultivated lands and regions with a trend of decreasing FVC may also indicate that competition for water with agricultural activities may have limited the recovery of the surrounding vegetation.

According to the hydrological records and satellite-based observations in our study, the implementation of EWDP since 2000 has stopped further ecological degradation that had been occurring since the 1980s and positively influenced the recovery of the ecosystems within the Ejina Oasis. The increased river discharge supported the vegetation recovery in the near-river

channel regions (within ~1 km) and maintained the terminal lake system. However, the agricultural development stimulated by the additional water resource might have obstructed further ecosystem recovery because large amounts of water have been consumed by the agriculture sector. Therefore, if managers wish to achieve a higher level of ecosystem recovery, the ecological water allocation schemes need to be maintained or enhanced. Also, regulations and practices such as "grain for green" and water-saving agriculture should be carried out within the downstream basin to ensure the allocated water flows into the

ecosystem.

**Acknowledgement** This work was funded by the Australian Research Council (Project No: ARC-FT130100274) and National Natural Science Foundation of China (No: 41301036).

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





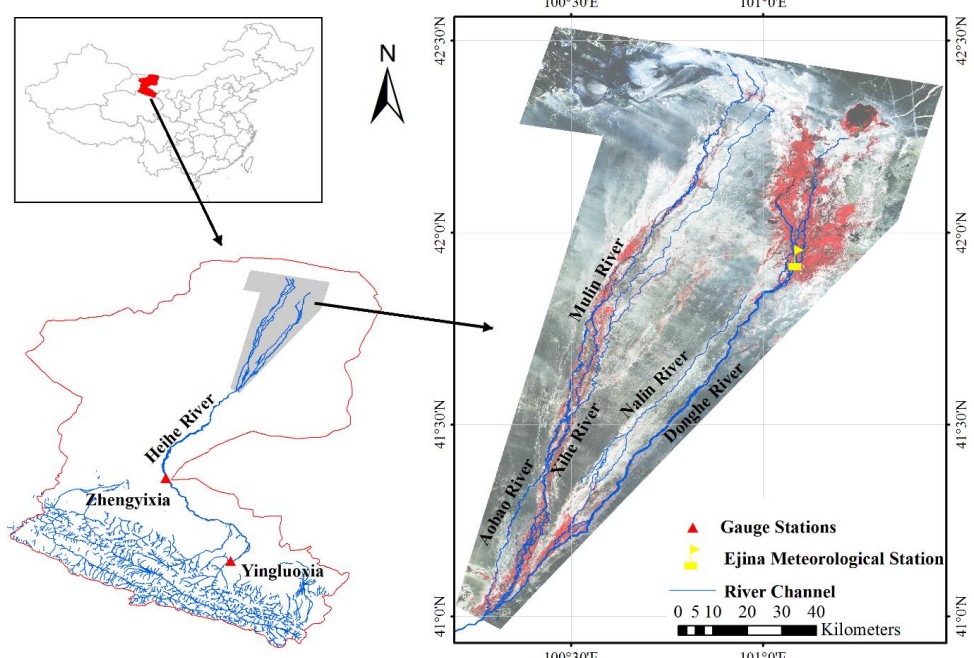

**Figure 1: Location of the Heihe River Basin and Ejina Oasis. The enlarged map is a Landsat image taken on 24 August 2011. Near infrared, red, and green bands are used to create this false-color image. Dark gray to white, desert-covered land dominates this region while red, plant-covered land major distributed along the river channels.**

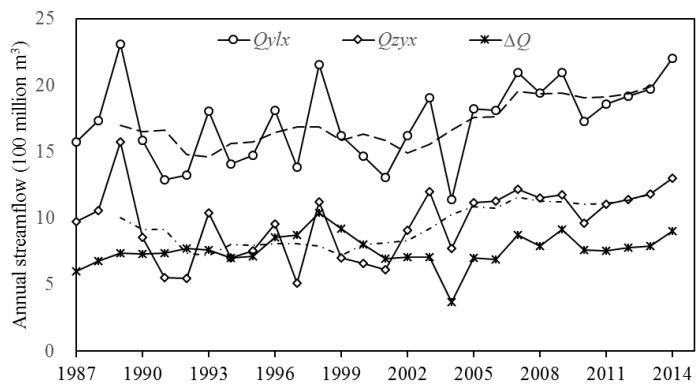

**Figure 2: Temporal variations of streamflow at YLX (circle) and ZYX (star) and the difference between the two (cross). The 5-year moving average annual discharge for YLX (dash) and ZYX (dash-dot) are presented to show changing trends.**





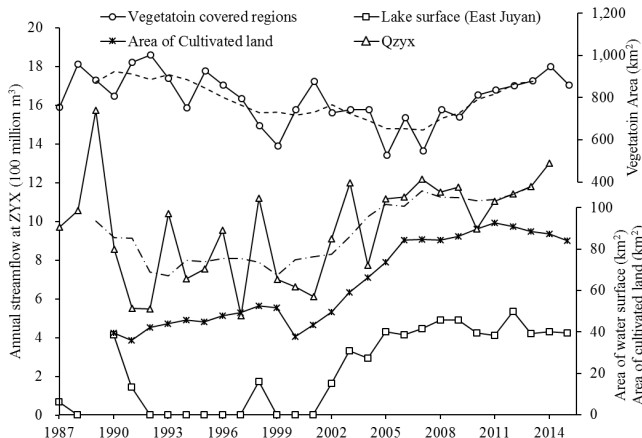

**Figure 3: Temporal variations in river discharge at ZYX (triangle), annual total vegetation covered regions (circle), East Juyan Lake surface area (square) and cultivated lands (cross) from 1987 to 2015. The dash line and dash-dot line denote the 5-year moving average values for vegetation areas and streamflow at ZYX respectively.**

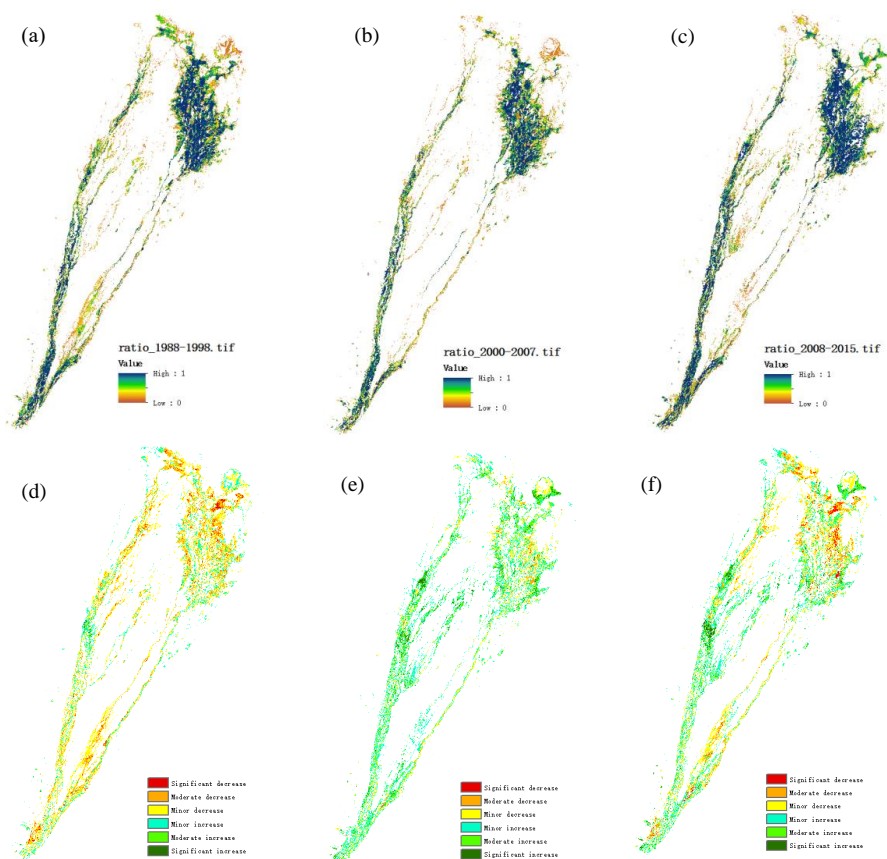

**Figure 4: Spatial distribution of vegetation covered regions in Ejina Oasis. The upper row shows the frequency of vegetation distribution for the three periods: (a) period 1 (1988–1998), (b) period 2 (2000–2007) and (c) period 3 (2008–2015). The lower row shows the corresponding changes between the pairs of periods: (c) changes between periods 1 and 2, (d) changes between periods 2 and 3 and (e) changes between periods 1 and 3.**





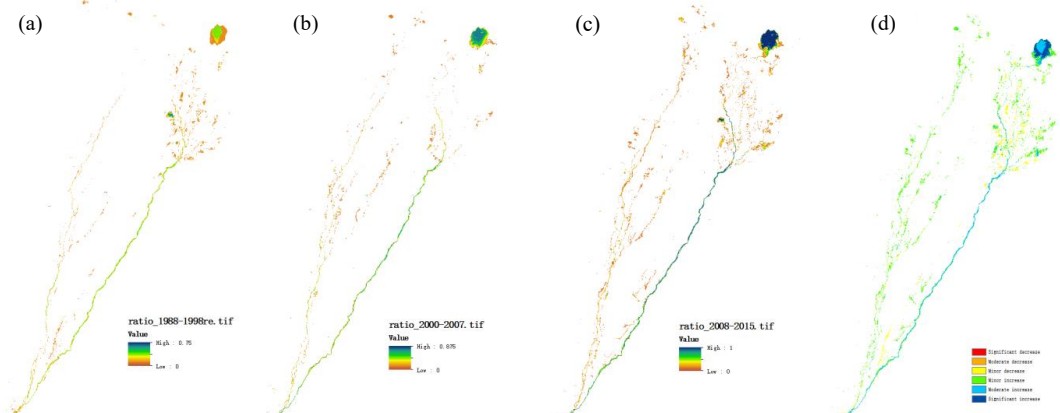

**Figure 5: Frequency of inundation for (a) period 1 (1988–1998), (b) period 2 (2000–2007) and (c) period 3 (2008–2015); and (d) the changes of rate between periods 1 and 3.**



**Table 1: Selected metrics for presenting ecosystem dynamics**

| Components | Metrics | Description |
|---|---|---|
| Water | Spatial extent | Regions with detectable water distribution, including inundated river channels, ponds and terminal lakes. |
| | Frequency index | Number of times a pixel is distributed with water for a specific period and divided by the length of the period (Thomas et al., 2011). |
| Vegetation (overall/100 m, 400 m, 1000 m buffer zones) | Spatial extent | Regions with detectable vegetation distribution. |
| | Frequency index | Number of times a pixel is distributed with vegetation and divided by the length of the period. |
| | Fractional vegetation cover | The fraction of green vegetation within a pixel. |
| Cultivated lands | Spatial extent | Identified regions with sign of agro-activities (e.g. regions with periodic high red and near infrared reflectance, and landscape with ridges) |

**Table 2: Linear regression models for determining the areas of vegetation cover, cultivated lands and terminal lakes. $Q_p$, $Q_c$ and $Q_{cp}$ stands for the annual discharge of previous ($p$), current ($c$) years and sum of previous and current years measured at ZYX station, respectively.**

| | Linear regression models |
|---|---|
| Vegetation distribution | Area (veg) = –18.718 × T + 2582.498 ($R^2 = 0.526$, $F = 20.01$, $p = 0.000$) |
| FVC | (no statistical significant relationship with hydrological and climatic variables was found) |
| Terminal lake area | Area (lake) = 2.516 × $Q_p$ + 5.800 ($R^2 = 0.365$, $F = 6.178$, $p = 0.038$) |
| | Area (lake) = 2.731× $Q_p$ +3.532 × $Q_c$ – 31.726 ($R^2 = 0.841$, $F = 18.545$, p = 0.002) |
| | Or Area (lake) = 2.974 × $Q_{cp}$ – 28.521 ($R^2 = 0.825$, $F = 37.746$, p = 0.000) |
| Cultivated land | Area (agri) = 5.19× $Q_c$ +8.689 ($R^2 = 0.530$, $F = 18.041$, $p = 0.001$) |

5 **Table 3: Pearson correlation coefficients indicating relationship between vegetation dynamics (spatial distribution and FVC) and variations in river flows. "–" means no significant correlation was detected while "*" and "**" denoted that the detected correlation was significant at 0.05 and 0.01 level, respectively.**

| Buffer zones | Regions | Area | | | | FVC | | | |
|---|---|---|---|---|---|---|---|---|---|
| | | $Q_c$ | $Q_{p3}$ | $Q_{p5}$ | $T$ | $Q_c$ | $Q_{p3}$ | $Q_{p5}$ | $T$ |
| 100 m | All regions | – | 0.496* | 0.703** | –0.472* | – | 0.545** | 0.601** | – |
| | Xihe River | – | 0.468* | 0.691* | – | – | – | – | – |
| | Donghe River | – | – | 0.455* | –0.471* | – | 0.571** | 0.663** | – |
| 400 m | All regions | – | – | 0.555** | –0.600** | – | 0.437* | 0.538** | – |
| | Xihe River | – | – | 0.634** | –0.572** | – | – | – | – |
| | Donghe River | – | – | – | –0.598** | – | 0.775** | 0.830** | – |
| 1000 m | All regions | – | – | 0.549** | –0.611** | – | 0.475* | 0.548** | – |
| | Xihe River | – | – | 0.639** | –0.594** | – | – | – | – |
| | Donghe River | – | – | – | –0.612** | – | 0.806** | 0.857** | – |

**Table 4: Change rates (%) within different buffer zones (coefficient of determination ($R^2$) indicated in parenthesis).**

| Buffer zone | Trend before 2003 | | | Trend after 2003 | | |
|---|---|---|---|---|---|---|
| | All regions | Xihe | Donghe | All regions | Xihe | Donghe |
| 100 m | –0.09(0.20) | –0.11(0.23) | –0.03(0.03) | 0.22(0.47) | 0.28(0.69) | 0.49(0.65) |
| 400 m | –0.19(0.50) | –0.24(0.59) | –0.1(0.20) | 0.25(0.58) | 0.17(0.40) | 0.36(0.69) |
| 1000 m | –0.21(0.54) | –0.26(0.60) | –0.15(0.30) | 0.2(0.47) | 0.11(0.19) | 0.33(0.72) |