# Peer review of "Downstream ecosystem responses to middle reach regulation of river discharge in the Heihe River Basin, China"

_Hydrology and Earth System Sciences, 2016_

## Referee Comment (RC1) · Anonymous Referee #1 · 22 Jul 2016

The paper used Landsat images to identify the spatial extent and fractional coverage of the oasis ecosystem and then used a linear model to analyze their relationship with hydrological and climatic variables in the Heihe River in China. Since the downstream oasis system is a typical ecosystem in the arid northwest China, the study would improve understanding the evolution of these oasis ecosystem and the effects of river regulations.

I'm not familiar with the Landsat images processing, but I think it should also be like some kind of model. I want to know if some calibration is done for the vegetation coverage using filed observations or measurements. How about the uncertainty of the calculation results?

The authors found that none of hydrological variables has significant influence on vegetation distribution, but regional temperature has significant negative effects. Does this mean that the regional evapotranspiration has great effect? Why you did not analysis the ET variations?

Some specific comments: Line 1 on Page 4: Where is the West Juyan Lake and East Juyan Lake, should be marked in Figure 1. Line 1 on Page 8: "(-0.33 to 0)", is it correct? Line 13 on Page 13: As no groundwater depth and groundwater flow information was presented, the authors should provide more solid evidence that groundwater depth is disturbed only in <300 m area away from the rivers. How you determined the number of 300 m? If the river water discharge to groundwater, groundwater depth would increase away from the river. Figures 4 and 5: The text size of the legend is too small.

---

## Referee Comment (RC2) · X. Li (Referee) · 23 Jul 2016

**General comments**

This work is an important contribution to the water resource management in arid regions. The authors carried out a detailed and reliable analysis on the changes in vegetation distribution, water bodies and cultivated lands from 1987 to 2015 in a typical inland river basin, the Heihe River Basin. Although this river basin has been well studied, the yearly variation of land cover and its relationship with water resource management at the whole river basin scale has never been quantified. This paper addressed this challenge by producing yearly high-resolution land cover and inundated area maps and analyzing the land cover change from some unique viewpoints. I particularly like the

following two points. (1) The relationship between vegetation change and streamflow in previous years (Qp3, Qp5, and sum of the streamflow of previous and current years). This lead to an interesting conclusion indicated by the authors "... repeated inflows in consecutive years are required to push water into the terminal lake, and hence support the surrounding environment". I believe this conclusion has important implications for water resource management in the terminal area of inland river basins: the ecosystem restoration is a long-term process in arid region. (2) Vegetation change along the river side using a buffer analysis is interesting (.. a series of buffer zones along river channels (100 m, 400 m and 1000 m away from river channel) were introduced to detect the interactions of vegetation dynamics and river flow). I have seen a lot of studies on the relationship between vegetation and streamflow. But this new analysis provide new insight. It reveals that the streamflow only has impact on vegetation distributed in the narrow belt along the river.

Generally, this paper can be accepted after a minor revision. I would suggest the authors to address the following issues in their revised manuscript.

(1) Change of cropland has a very significant impact on the hydrological cycle in the Ejina oasis area and might lead to some policy change on water diversion in upstream. I would suggest that the authors to provide an in-depth analysis of the impact of cropland expansion on water resource management, not only at the downstream area but also on the whole river basin. For example, 0.1-0.2*10^9 m^3 yr^-1 of water might have been used by cropland. According to the water diversion agreement (EWDP) of the HRB, all the cropland in the downstream area should be converted to natural vegetation. Therefore, cropland reclamation in the downstream area is somehow illegal. If the cropland can be appropriately managed, I believe the ecosystem restoration in the downstream area of the HRB will be more significant. I would suggest the authors to expand the discussion in the first paragraph of Page 13.

(2) Also for the cropland, I would suggest the authors to compare their mapping results with that by Hu et al., 2015 (Hu, X. L., L. Lu, X. Li, J. H. Wang, and X. G. Lu. 2015. Ejin

Oasis land use and vegetation change between 2000 and 2011: the role of the Ecological Water Diversion Project. Energies 8:7040-7057). An evaluation of the accuracy is suggested to be added to Section 2.4.2.

(3) P7, second paragraph. Actually, some FVC datasets for the Heihe River Basin have been produced (1km: http://westdc.westgis.ac.cn/data/21d993d3-841a-4d04-9647-82c21601a739 and 30m: http://westdc.westgis.ac.cn/data/aa9f7e76-363f-4e21-b44c-adf50dd96b0f). I would suggest the authors to use the datasets in their analysis or validation of FVC products.

Specific comments:

P2, L 28, "since the establishment of new China government in 1949, the Basin has experienced water and ecological stress" –> "since the quick population growth in 1940s, the HRB has experienced water and ecological stress".

P3, L13-14, "To our knowledge, this is the first attempt to apply high-resolution images and long term datasets in Ejina Oasis" –> "To our knowledge, this is the first attempt to apply long term high-resolution remote sensing derived land cover datasets in Ejina Oasis".

P4, L22, add a reference. Li, X., Z. T. Nan, G. D. Cheng, Y. J. Ding, L. Z. Wu, L. X. Wang, J. Wang, Y. H. Ran, H. X. Li, X. D. Pan, and Z. M. Zhu. 2011. Toward an improved data stewardship and service for environmental and ecological science data in west China. International Journal of Digital Earth 4:347-359.

P12, L11, "data not presented here". Why?

References

Lu, L., X. Li, and G. D. Cheng. 2003. Landscape evolution in the middle Heihe River Basin of northwest China during the last decade. Journal of Arid Environments 53:395-408.

Wang, G. X., J. Q. Liu, J. Kubota, L. Chen, G. AF Wang, J. Liu, J. Kubota, and L. Chen. 2007. Effects of land-use changes on hydrological processes in the middle basin of the Heihe River, northwest China. Hydrological Processes 21:1370-1382.

---

## Author Comment (AC1) · 5 Sep 2016

Referee #1's comments 1: I'm not familiar with the Landsat images processing, but I think it should also be like some kind of model. I want to know if some calibration is done for the vegetation coverage using filed observations or measurements. How about the uncertainty of the calculation results?

Authors' response: Thanks for this valuable comment. A similar comment was given by Referee #2. We carried out several straight-forward processing when we prepared the Landsat: converting the original Digital Numbers into reflectance at top of atmosphere and then calibrating it into surface reflectance through an atmospheric correction. The algorithms applied during these procedures were based on either the parameters provided within the metadata of the image or well-established methods in this field. We don't need to validate the band reflectance at this stage, but we do need to validate the parameters derived from the Landsat images. In the submitted version of the manuscript, we discussed our ways of validating the classification maps and FVC results, including comparing with high resolution satellite images (for validating classification maps) and checking FVC values in desert areas. In addition, as our response to Reviewer #2 shows, we will use existing land use and land cover maps to validate our results and use field measured FVC values to evaluate the accuracy of our FVC estimations. We will revise our manuscript accordingly.

Referee #1's comments 2: The authors found that none of hydrological variables has significant influence on vegetation distribution, but regional temperature has significant negative effects. Does this mean that the regional evapotranspiration has great effect? Why you did not analysis the ET variations?

Authors' response: Thanks for this comment. We agree with Referee #1 that increasing temperature indicated higher ET levels. However, landscape in areas away from river channel in the arid Ejina region was dominated with desert and Gobi, and annual ET in Ejina was extremely low, usually less than 50 mm (Lian et al., 2015), some studies even found that daily ET in Ejina was less than 1 mm/d in its surrounding areas during wet seasons (Luo et al., 2012). Only in cropping and natural vegetation regions show slight higher ET levels. Streamflow and streamflow recharged groundwater were the major water sources supporting vegetation development, therefore, we focused on these datasets and didn't analyze the impacts of ET variations.

Additional reference:

Luo, X., Wang, K., Jiang, H., Sun, J. and Zhu, Q. Estimation of land surface evapotranspiration over the Heihe River basin based on the revised three-temperature model, Hydrological Processes, 26, 1263-1269, 2012.

Lian, J. and Huang, M. Evapotranspiration Estimation for an Oasis Area in the Heihe

River Basin Using Landsat-8 Images and the METRIC Model, Water Resour. Manage., 29, 5157-5170, 2015.

Referee #1's comments 3: Some specific comments: (listed below (in bold))

Line 1 on Page 4: Where is the West Juyan Lake and East Juyan Lake, should be marked in Figure 1.

Authors' response: We will update figure 1 with the location of the West and East Juyan Lake.

Line 1 on Page 8: "(-0.33 to 0)", is it correct?

Authors' response: (-0.33 to 0) was a mistake in the current version, we will correct it as noted.

Line 13 on Page 13: As no groundwater depth and groundwater flow information was presented, the authors should provide more solid evidence that groundwater depth is disturbed only in <300 m area away from the rivers. How you determined the number of 300 m? If the river water discharge to groundwater, groundwater depth would increase away from the river.

Authors' response: We collected groundwater observation data from the WestDC database in which groundwater levels were measured at 50 m, 300 m, 2,200 m, 2,700 m, 3,200 m and 4,300 m away from the river channel along a transection located in the study area as shown in the following figure (Fig.1). We will include this figure as well as some details of the sampling points (latitude and longitude, elevation, sampling periods.) in our revised manuscript.

Figures 4 and 5: The text size of the legend is too small.

Authors' response: we will modify the figures when we revise the manuscript.

[Figure]

**Fig. 1.** Groundwater variations along a transection in Ejina Oasis. Solid line with circles indicates mean levels, dash lines indicate minimum and maximum water levels.

---

## Author Comment (AC2) · 5 Sep 2016

Prof. Xin Li's comments 1: Change of cropland has a very significant impact on the hydrological cycle in the Ejina oasis area and might lead to some policy change on water diversion in upstream. I would suggest that the authors to provide an in-depth analysis of the impact of cropland expansion on water resource management, not only at the downstream area but also on the whole river basin. For example, $0.1-0.2*10^9$ m$^3$ yr$^{-1}$ of water might have been used by cropland. According to the water diversion agreement (EWDP) of the HRB, all the cropland in the downstream area should be converted to natural vegetation. Therefore, cropland reclamation in the downstream area is somehow illegal. If the cropland can be appropriately managed, I believe the

ecosystem restoration in the downstream area of the HRB will be more significant. I would suggest the authors to expand the discussion in the first paragraph of Page 13.

Authors' response: We fully agree with Prof. Li's concern about the impacts of cropland expansion on water cycle in the Ejina Oasis. Prof. Li's comments of "appropriately manage local cropland could promote downstream restoration" supported our argument of "the agricultural development stimulated by the additional water resource might have obstructed further ecosystem recovery because large amounts of water have been consumed by the agriculture sector". However, given that the objective of this article was to understand the downstream ecosystem responses to middle reach regulation of river discharge, we analyzed the cropland distribution in the downstream to help understanding the competition relationship between crop and natural vegetation on water use. Therefore, we didn't expand our study into the crop land in the middle stream of HRB. As a matter of fact, the impact of middle stream agricultural development on the downstream ecosystem has been embedded in the streamflow variations flowing into the down stream, since agriculture in the middle stream of HRB is the major water consumer.

Prof. Xin Li's comments 2: Also for the cropland, I would suggest the authors to compare their mapping results with that by Hu et al., 2015 (Hu, X. L., L. Lu, X. Li, J. H. Wang, and X. G. Lu. 2015. Ejin Oasis land use and vegetation change between 2000 and 2011: the role of the Ecological Water Diversion Project. Energies 8:7040-7057). An evaluation of the accuracy is suggested to be added to Section 2.4.2.

Authors' response: We appreciate Prof. Li's suggestion and compared our results with Hu's results in 2000 and 2011 (Hu et al., 2015). Our results show high consistency with Hu's results as displayed in Fig.1 (our results in green and Hu's results in red). We will revise the relevant part of our manuscript with the comparison results from this new dataset. An evaluation of the accuracy would be added to Section 2.4.2.

Accordingly, we will also revise the "Acknowledgement" section to deliver our thanks to

Dr Xiaoli Hu for sharing us with her land use and land cover maps in 2000 and 2011. The reference list will be updated as well.

Prof. Xin Li's comments 3: P7, second paragraph. Actually, some FVC datasets for the Heihe River Basin have been produced (1km: http://westdc.westgis.ac.cn/data/21d993d3-841a-4d04- 9647-82c21601a739 and 30m: http://westdc.westgis.ac.cn/data/aa9f7e76-363f-4e21- b44c-adf50dd96b0f). I would suggest the authors to use the datasets in their analysis or validation of FVC products.

Authors' response: Once again, we appreciate that Prof. Li provided data sources for us to improve our results quality. We related our 2014 FVC estimation with the field data collected in 2014 in (http://westdc.westgis.ac.cn/data/c008ca1d-dd30-44cc-876d-9298dd07982d) and found that, the FVC values derived from Landsat images presented relatively high accuracy as indicated in the following figure (Fig.2). We will revise the relevant part of our manuscript with the comparison results from this new dataset accordingly. A map showing location of the field sampling points as well as some details on sampling methods will be also added in a supplementary document.

Reference : Observation Dataset of fractional vegetation cover by digital camera in the lower reaches of the Heihe River Basin. Cold and Arid Regions Environmental and Engineering Research Institute, Chinese Academy of Sciences. 2015. 10.3972/hiwater.271.2015.db)

Prof. Xin Li's specific comments: (see paragraphs (in bold) listed below)

P2, L 28, "since the establishment of new China government in 1949, the Basin has experienced water and ecological stress" ->"since the quick population growth in 1940s, the HRB has experienced water and ecological stress".

Authors' response: We agree and will modify the sentence as suggested.

P3, L13-14, "To our knowledge, this is the first attempt to apply high-resolution images

and long term datasets in Ejina Oasis" -> "To our knowledge, this is the first attempt to apply long term high-resolution remote sensing derived land cover datasets in Ejina Oasis".

Authors' response: we agree and will update the sentence accordingly.

P4, L22, add a reference. Li, X., Z. T. Nan, G. D. Cheng, Y. J. Ding, L. Z. Wu, L.X. Wang, J. Wang, Y. H. Ran, H. X. Li, X. D. Pan, and Z. M. Zhu. 2011. Toward an improved data stewardship and service for environmental and ecological science data in west China. International Journal of Digital Earth 4:347-359.

Authors' response: We will update our reference list.

P12, L11, "data not presented here". Why?

Authors' response: the description of "Observations in this area (data not presented here) have found that the annual average groundwater depth alongside the river (< 300 m) was around 2 m, which increased to more than 3.5 m in remote regions (> 4300 m away from river channels)." was derived from the long term groundwater level observation in downstream HRB, which was also archived in the WestDC database. We will add a table or a figure showing the observed groundwater variations when we revise our manuscript to provide readers with more detailed information.

References

Lu, L., X. Li, and G. D. Cheng. 2003. Landscape evolution in the middle Heihe River Basin of northwest China during the last decade. Journal of Arid Environments 53:395-408.

Wang, G. X., J. Q. Liu, J. Kubota, L. Chen, G. AF Wang, J. Liu, J. Kubota, and L. Chen. 2007. Effects of land-use changes on hydrological processes in the middle basin of the Heihe River, northwest China. Hydrological Processes 21:1370-1382.

[Figure]

**Fig. 1.** Comparison of cropland distribution maps in 2011 (left to right, our overall results, Hu's overall results, our results in major cropping region and Hu's results in major cropping area)

[Figure]

[Figure]

**Fig. 2.** FVC validation with field measurements (right panel indicated location of plots)

---

## Author Response (AR1)

**Responses to Anonymous referee #1's comments**

| Referee' comments | Authors' response |
|---|---|
| **Referee #1's comments 1**: I'm not familiar with the Landsat images processing, but I think it should also be like some kind of model. I want to know if some calibration is done for the vegetation coverage using filed observations or measurements. How about the uncertainty of the calculation results? | Thanks for this valuable comment. A similar comment was given by Referee #2. We carried out several straight-forward processing when we prepared the Landsat: converting the original Digital Numbers into reflectance at top of atmosphere and then calibrating it into surface reflectance through an atmospheric correction. The algorithms applied during these procedures were based on either the parameters provided within the metadata of the image or well-established methods in this field.

We don't need to validate the band reflectance at this stage, but we do need to validate the parameters derived from the Landsat images. In the submitted version of the manuscript, we discussed our ways of validating the classification maps and *FVC* results, including comparing with high resolution satellite images (for validating classification maps) and checking *FVC* values in desert areas. To further address this point, we collected existing land use maps and field measured *FVC* values to assess the accuracy of the results, some results were added to section 2.4.3 "Validation of derived metrics" as described below:

" *...Accuracy of the established maps was further assessed with existing land cover maps in 2000 and 2011 (Hu et al., 2015). The inter-comparison found that the two datasets presented substantial consistency where kappa coefficients (k) were 0.7206 and 0.6731 for 2000 and 2011, respectively. Details about the comparison and the confusion matrix were provided in Figure S1 and Table S1.*
*To verify the FVC results, we related our 2014 FVC estimation to 54 field-measured FVC values which were available in the WestDC database (Wang et al., 2015). There was a good agreement between calculated and field measurements, with a high correlation (Figure 2, $R^2 = 0.92$, p-value < 0.001)....* "

In addition, details about the procedures validating our vegetation and crop distribution maps were organized in a supplementary document. A flowchart as well as the confusion matrix presenting the accuracy were provided. Please turn to the uploaded supplementary material for details. |

| | |
|---|---|
| **Referee #1's comments 2:** The authors found that none of hydrological variables has significant influence on vegetation distribution, but regional temperature has significant negative effects. Does this mean that the regional evapotranspiration has great effect? Why you did not analysis the ET variations? | Thanks for this comment. We agree with Referee #1 that increasing temperature indicated higher ET levels. However, landscape in areas away from river channel in the arid Ejina region was dominated with desert and Gobi, and annual ET in Ejina was extremely low, usually less than 50 mm (Lian et al., 2015), some studies even found that daily ET in Ejina was less than 1 mm/d in its surrounding areas during wet seasons (Luo et al., 2012). Only in cropping and natural vegetation regions show slight higher ET levels. Streamflow and streamflow recharged groundwater were the major water sources supporting vegetation development, therefore, we focused on these datasets and didn't analyze the impacts of ET variations. To address this point, we modified section 2.2 "Hydrological and climatic variables influencing the downstream ecosystems" to clarify our criteria in selecting the metrics as detailed below:

*"… Since the landscape in lower HRB was dominated with desert and Gobi, annual ET in Ejina was usually less than 50 mm (Lian et al., 2015). Only regions near the river showed relatively higher ET levels but they were also supported by the river flows (Luo et al., 2012). Therefore, we excluded ET in this analysis. Furthermore, although groundwater observations covering the entire study period were not available, short term measurements (2010 - 2012) at a transection located in the Ejina Oasis (Figure 1) were collected from WestDC to discuss the impact of groundwater on the downstream vegetation dynamics…"*

The reference list was updated accordingly. |
| **Referee #1's specific comments:**
**L1P4:** Where is the West Juyan Lake and East Juyan Lake, should be marked in Figure 1.

**L1P8:** "(-0.33 to 0)", is it correct?

**L13P13:** As no groundwater depth and groundwater flow information was presented, the authors should | We appreciate the referee kindly pointed out the problems in presenting the results, we have modified the figures and sentences accordingly as detailed below:

**L1P4:** We updated Figure 1 with the following elements, please turn to Figure 1 for details:
✓ Location and spatial extent (Maximum level) of West and East Juyan Lake
✓ FVC sampling points
✓ Groundwater observation location |

| | |
|---|---|
| provide more solid evidence that groundwater depth is disturbed only in <300 m area away from the rivers. How you determined the number of 300 m? If the river water discharge to groundwater, groundwater depth would increase away from the river.

**Figures 4 and 5:** The text size of the legend is too small. | **L1P8:** We have corrected the (-033 to 0) to (-0.33 to 0) in the article.

**L13P13:** We collected groundwater observation data from the WestDC database in which groundwater levels were measured at 50 m, 300 m, 2,200 m, 2,700 m, 3,200 m and 4,300 m away from the river channel along a transection located in the study area. To clarify this point, we marked the location of the transection in Figure 1 and a detailed locations of the sampling points were provided in the "Supplementary material" (Figure S2). Figure S2 also presented groundwater variations along the transection.

**Figures 4 and 5:** we have updated text size of the legends in Figure 4 and Figure 5. |
| **Responses to Prof. Xin Li's comments** | |
| **Prof. Xin Li's comments 1:** Change of cropland has a very significant impact on the hydrological cycle in the Ejina oasis area and might lead to some policy change on water diversion in upstream. I would suggest that the authors to provide an in-depth analysis of the impact of cropland expansion on water resource management, not only at the downstream area but also on the whole river basin. For example, 0.1-0.2*10^9 m^3 yr^-1 of water might have been used by cropland. According to the water diversion agreement (EWDP) of the HRB, all the cropland in the downstream area should be converted to natural vegetation. Therefore, cropland reclamation in the downstream area is somehow illegal. If the cropland can be appropriately managed, I believe the ecosystem restoration in the downstream area of the HRB will be more significant. | We fully agree with Prof. Li's concern about the impacts of cropland expansion on water cycle in the Ejina Oasis. Prof. Li's comments of "appropriately manage local cropland could promote downstream restoration" supported our argument of "*the agricultural development stimulated by the additional water resource might have obstructed further ecosystem recovery because large amounts of water have been consumed by the agriculture sector*". However, given that the objective of this article was to understand the downstream ecosystem responses to middle reach regulation of river discharge, we analyzed the cropland distribution in the downstream to help understanding the competition relationship between crop and natural vegetation on water use. Therefore, we didn't expand our study into the crop land in the middle stream of HRB. As a matter of fact, the impact of middle stream agricultural development on the downstream ecosystem has been embedded in the streamflow variations flowing into the downstream, since agriculture in the middle stream of HRB is the major water consumer. |

| | |
|---|---|
| I would suggest the authors to expand the discussion in the first paragraph of Page 13. | |
| **Prof. Xin Li's comments 2:** Also for the cropland, I would suggest the authors to compare their mapping results with that by Hu et al., 2015 (Hu, X. L., L. Lu, X. Li, J. H. Wang, and X. G. Lu. 2015. Ejin Oasis land use and vegetation change between 2000 and 2011: the role of the Ecological Water Diversion Project. Energies 8:7040-7057). An evaluation of the accuracy is suggested to be added to Section 2.4.2. | We appreciate Prof. Li's suggestion and compared our results with Hu's results in 2000 and 2011 (Hu et al., 2015). Our results show high agreement with Hu's results. Description of the accuracy evaluation results was added to the section 2.4.2 and details already listed in our response to "**Referee #1's comments 1**".

The "Supplementary materials" also included the methods adopted for validating land cover maps and the results of accuracy evaluation. |
| **Prof. Xin Li's comments 3:** P7, second paragraph. Actually, some FVC datasets for the Heihe River Basin have been produced (1km: http://westdc.westgis.ac.cn/data/21d993d3-841a-4d04-9647-82c21601a739 and 30m: http://westdc.westgis.ac.cn/data/aa9f7e76-363f-4e21-b44c-adf50dd96b0f). I would suggest the authors to use the datasets in their analysis or validation of FVC products. | Once again, we appreciate that Prof. Li provided data sources for us to improve our results quality. We related our 2014 *FVC* estimation with the field data collected in 2014 in (http://westdc.westgis.ac.cn/data/c008ca1d-dd30-44cc-876d-9298dd07982d) and found that, the *FVC* values derived from Landsat images presented relatively high accuracy as indicated in Figure 2. We modified Section 2.4.2 with the new derived evidence as described below:

"…*To verify the FVC results, we related our 2014 FVC estimation to 54 field-measured FVC values which were available in the WestDC database (Wang et al., 2015). There was a good agreement between calculated and field measurements, with a high correlation (Figure 2, $R^2$ = 0.92, p-value < 0.001) …*"

The location of the *FVC* sampling sites were also indicated in Figure 1. |
| **Prof. Xin Li's specific comments:**

**P2L28:** "since the establishment of new China government in 1949, the Basin has experienced water and ecological stress" → "since the quick population growth in 1940s, the HRB has experienced water and ecological stress".

**P3L13-14:** "To our knowledge, this is the first attempt to apply high-resolution images and long term datasets | We thank Prof. Li for pointing out the problems in English presentation and providing suggestions to revise the sentences. We modified the parts accordingly as detailed below:

**P2L28:** We modified the sentence as suggested.

**P3L13-14:** We modified the sentence accordingly.

**P4L22:** We checked the reference and updated the reference list as suggested

**P12L11:** The description of "Observations in this area (data not presented here) have found that the annual average groundwater depth alongside the river (< 300 m) was around 2 m, which |

in Ejina Oasis" → "To our knowledge, this is the first attempt to apply long term high-resolution remote sensing derived land cover datasets in Ejina Oasis".

**P4L22:** add a reference. Li, X., Z. T. Nan, G. D. Cheng, Y. J. Ding, L. Z. Wu, L.X. Wang, J. Wang, Y. H. Ran, H. X. Li, X. D. Pan, and Z. M. Zhu. 2011. Toward an improved data stewardship and service for environmental and ecological science data in west China. International Journal of Digital Earth 4:347-359.

**P12L11:** "data not presented here". Why?

increased to more than 3.5 m in remote regions (> 4300 m away from river channels)." was derived from the long term groundwater level observation in downstream HRB, which was also archived in the WestDC database. To clarify this point, we marked the location of the sampling points in both Figure 1 and Figure S2, and groundwater variations at the sites were provided in Figure S2 as well.

[revised manuscript text omitted]
 | –0.21(0.54) | –0.26(0.60) | –0.15(0.30) | 0.2(0.47) | 0.11(0.19) | 0.33(0.72) |